# Membrane Distillation for the Production of Pharmaceutical-Grade Water—Investigation into the Application of AGMD and VMD

**DOI:** 10.3390/ijerph18116058

**Published:** 2021-06-04

**Authors:** Cornelius Nellessen, Thomas Klein, Hans-Jürgen Rapp, Frank Rögener

**Affiliations:** 1Wilhelm Werner GmbH, 51381 Leverkusen, Germany; cornelius.nellessen@werner-gmbh.com (C.N.); thomas.klein@werner-gmbh.com (T.K.); 2Institute of Chemical Process Engineering and Plant Design, Technische Hochschule Köln, 50679 Köln, Germany; 3Deukum GmbH, 72636 Frickenhausen, Germany; info@deukum.de

**Keywords:** membrane distillation, pharmaceutical-grade water, pure water, AGMD, VMD

## Abstract

The production of pharmaceutical ingredients, intermediates and final products strongly depends on the utilization of water. Water is also required for the purification and preparation of reagents. Each specific application determines the respective water quality. In the European Union, the European Pharmacopeia (Ph. Eur.) contains the official standards that assure quality control of pharmaceutical products during their life cycle. According to this, the production of water for pharmaceutical use is mainly based on multi-stage distillation and membrane processes, especially, reverse osmosis. Membrane distillation (MD) could be an alternative process to these classical methods. It offers advantages in terms of energy demand and a compact apparatus design. In the following study, the preparation of pharmaceutical-grade water from tap water in a one-step process using MD is presented. Special emphasis is placed on the performance of two different module designs and on the selection of optimum process parameters.

## 1. Introduction

The forecast of the European pharmaceutical market predicts a 3.9% growth between 2019 and 2024, while the global market is expected to rise by about 4.2% in the same period [1]. For the production of pharmaceutical ingredients, intermediates, and final products, as well as for the purification and preparation of reagents, water is crucial. However, water used for pharmaceutical applications is subject to strict regulations. In the European Union, the European Pharmacopeia (Ph. Eur) determines the quality of water for pharmaceutical use as well as the processing measures. 

Within the current pharmacopoeias, a distinction is made between purified water (PW) and water for injection (WFI) [2]. PW is used for the production of medical products that are neither pyrogen-free nor sterile. Pyrogens are substances that can cause fever in humans during parenteral intake (bypassing the intestine, e.g., intravenously). WFI is water for the production of medical products, solutions and dilutions for parenteral use. Drinking water with a quality according to the respective national regulations is the raw material for both PW and WFI production [2]. The Ph. Eur. sets limits and requirements for the quality of PW and WFI; some quality parameters are depicted in Table 1.

PW is produced by distillation, reverse osmosis (RO) in combination with electrodeionisation (EDI), ion exchange or other suitable methods [3]. In contrast, the Ph.Eur. significantly limits the processes that may be applied for the production of WFI. The most commonly used method for WFI generation is multi-stage pressure column distillation [3].

Membrane distillation (MD) may be an alternative process for the generation of pharma-grade water. MD is a thermal separation process based on vapor pressure differences between the feed and the distillate sides of porous, hydrophobic membranes [4]. When drinking water is used as a feed solution, these membranes allow the passage of water vapor only and retain suspended and dissolved matter on the retentate side; thus, the condensed product obtained is theoretically 100% pure water [5,6].

MD is investigated worldwide as a low-cost, energy-saving alternative to conventional separation processes such as distillation and reverse osmosis (RO). MD can be operated at a relatively low feed temperature between 50 °C and 80 °C. 

Some studies [7,8,9] employed newly developed single-layer and multilayer graphene-based membranes for application in membrane distillation. Xu et al. [7] reported about graphene oxide (GO) nanolayers that were deposited on the permeate side of PVDF membranes. Generally, these membranes show an advantage for membrane distillation because of their unique water channels. Application in direct-contact membrane distillation enhanced the desalination performance due to a decrease of the vapor pressure at the permeate side. Salt rejection was improved to 99.9%. Deterioration of the permeate quality caused by membrane wetting was avoided by the properties of the GO layer. Grasso et al. [8] prepared porous composite membranes from functionalized PVDF membranes that were coated with graphene. The generated membrane was applied in direct-contact membrane distillation (DCMD) and showed long-lasting salt rejections >99.9%. Huang et al. [9] reported about photothermal membrane distillation (PMD) using a PTFE membrane coated with an ultrathin graphene-based film. Compared to the unmodified membrane, the transmembrane water flux of the modified PTFE membranes showed an enhancement of about 78% under solar illumination.

For the operation of MD, it is important that the dry pores are not wetted by the liquid feed, which is directly in contact with the membrane. In contrast to RO, MD is not limited by the osmotic pressure that is generated; furthermore, equipment costs are lower [10]. As MD is operated at low temperature levels, the use of waste heat [11] as well as solar thermal energy and geothermal energy is possible [12,13]. On the other hand, a disadvantage of MD is the low permeate flow compared to RO [7]. This restriction may be overcome by the further development of graphene-based membranes. 

MD processes can be categorized into four basic module configurations, which play a fundamental role in separation efficiency and processing costs: direct-contact membrane distillation (DCMD), air-gap membrane distillation (AGMD), sweeping-gas membrane distillation (SGMD), and vacuum membrane distillation (VMD), as shown in Table 2. Furthermore, alternative configurations with low energy consumption and improved flux have been developed [14].

There is hardly any information available on the application of MD for the generation of pharma-grade water. A study [16] reported about the application of membrane distillation for the generation of WFI without intermediate steps. Data on the performance were not provided.

## 2. Materials and Methods

### 2.1. Test Plant and Modules

The investigations were carried out using a test plant that was especially designed by Wilhelm Werner GmbH (Figure 1). It allows the operation of both AGMD and VMD configurations, which were chosen for the experimental investigations due to their advantages, i.e., little conductive heat losses and relatively high permeate flux.

Generally, the plant was designed for testing the suitability of MD for the generation of pharma-grade water from tap water. The design criteria considered neither hygienic design nor energetic optimization. Hauser [17] stated that conventional detachable connections need to be replaced by solid connections or by hygienically designed detachable connections and sealing points to prevent the growth of microbiological impurities.

Figure 2 depicts the piping and instrumentation diagram of the two chosen configurations.

Deukum GmbH, Germany, provided a plate-and-frame membrane module Type MDX100 with membrane packages for AGMD and VMD, as shown in Figure 3. The module was operated in a counterflow mode, i.e., the feed and distillate flew along the membrane in different directions. The velocity of the respective solutions determines the thickness of the fluid dynamic boundary layer, which creates a resistance to material and heat transport. The membrane package used for AGMD consisted of the flat membranes (polyethylene), spacers (silicone) and condenser sheets (PP). The membrane package used for VMD consisted of the flat membranes (polyethylene) and spacers (silicone). Each membrane package was sealed with silicone, and the individual components were bonded together by heat treatment.

In total, 30 microporous polyethylene membranes type 14P02F from Lydall Performance Materials, each with an effective membrane area of 0.01 m^2^, were stacked in each module, leading to a total membrane area of 0.3 m^2^. The membrane characteristics are reported in Table 3.

### 2.2. Feed Solution and Cooling Water

Untreated tap water provided by the city of Leverkusen, Germany, served as feed solution. Tap water served as a cooling agent for the vapor condensation as well.

### 2.3. Conductivity Measurement

The electrical conductivity is an important quality parameter for the separation efficiency in the MD process. Conductivity was determined with a conductivity meter type JUMO Blackline CR-EC built into the pipeline.

### 2.4. Quality Parameters

The central factor for the efficiency assessment of MD is the transmembrane distillate flow, represented as mass flow or volume flow [18]. To compare the performance of different module or plant concepts, the distillate flow can be related to the efficient membrane surface A_M_. The resulting value is referred to as flux.

The degree of desalination is described by the salt retention *R*, which can be calculated using the conductivity of distillate *σ_D_* and feed *σ_F_* [19]:(1)$$R=\left(1-\frac{\sigma_{D}}{\sigma_{F}}\right)\cdot 100\%$$

The rejection of impurities of all kinds can be described by the decontamination factor *DF* [20]:(2)$$DF=\frac{c_{D}}{c_{F}}\cdot 100\%$$

In batchwise investigations, the yield *Y* describes the ratio of the generated distillate volume to the feed volume:(3)$$Y=\frac{V_{D}}{V_{F}}\cdot 100\%$$

### 2.5. Experimental Investigations

The investigations included factor variation of influencing parameters to determine optimum operating conditions and long-term experiments. Furthermore, investigations concerning hygienic safety of the process were conducted.

Before the examinations were carried out, the statistical variation of the distillate flux was tested at different working points. Statistical variation was so low that each test presented in our paper was carried out only once.

#### 2.5.1. Investigation of Influencing Parameters by Factor Variation

Full factorial parameter variations of feed temperature T_F_, feed volume flow F_F_ and cooling water volume flow F_K_, each on three factor levels, were performed. For VMD, also the applied negative pressure was varied. The cooling water temperature in all tests was kept constant at approximately 17.5 °C. The designations 1, 2 and 3 in Section 3 refer to the factor levels Low (1), Medium (2) and High (3). Furthermore, a reference value (0) is given for VMD, indicating operation without negative pressure.

Table 4 summarizes the experimental conditions.

#### 2.5.2. Long-Term Investigations

Long-term investigations were conducted with both AGMD and VMD configurations of the test plant. The generated distillate was drained constantly, while the concentrate was fed back to the feed tank. Thus, the concentrate concentration increased over time. Table 5 depicts the parameters that were chosen for these investigations. A higher cooling water volume flow could not be applied, as previous experiments showed that in this case, the feed temperature could not be kept stable for a sufficiently long time. Furthermore, it turned out that in the VMD configuration of the test plant, the feed tank contracted at a pressure below −0.15 bars. Since the employed vacuum pump could not be monitored automatically, the duration of the VMD investigation had to be restricted to 10 h for safety reasons.

#### 2.5.3. Hygienic Safety

For the determination of the total bacteria count in the distillate, the plant was first sterilized for 10 min using a 2 wt% hydrogen peroxide solution. Afterwards, the peroxide solution was rinsed out, and the plant was operated for 24 h at constant operating conditions prior to sampling of the feed solution and distillate. Table 6 indicates the operating conditions.

The microbiological investigations took place in an external lab. The total bacteria count of the feed water was measured according to the Germen Trinkwasserverordnung TVO based on DIN EN ISO 6222. The total bacteria count of the distillate was measured according to TVO and Ph. Eur. The examination based on the Ph. Eur. served to check whether the distillate met the requirements for PW or WFI according to Ph. Eur. The decontamination factor DF was calculated using the data of distillate and feed water determined according to TVO.

#### 2.5.4. Chemical Resistance of the Membranes against H_2_O_2_

The influence of the peroxide solution on spacer, condenser sheet and hydrophobicity of the applied membrane was examined. All materials were inserted for 24 h in a 2 wt% H_2_O_2_ solution, rinsed and then dried. Spacer and condensation film were visually inspected. A change of the contact angle of the membrane indicates whether the sterilization solution negatively affects the necessary hydrophobicity of the membrane. Thus, a drop test with distilled water was performed on untreated and treated membranes. For this, distilled water was dripped to the membrane sample with a pipette. The contact angle of the drop with the respective membrane sample was determined graphically.

## 3. Results

### 3.1. Full Factorial Parameter Variation for the Investigation of Optimum Operating Parameters

#### 3.1.1. AGMD

Figure 4, Figure 5 and Figure 6 show the results of the full factorial parameter variation.

Figure 4 depicts the influence of the feed temperature on the transmembrane distillate flux. Generally, the distillate flux increases linearly as the temperature rises.

The influence of the feed volume flow on the transmembrane distillate flux is depicted in Figure 5.

It is apparent that generally, the distillate flux increased with increasing feed volume flow, but the influence of crossflow conditions in the feed chamber on the distillate flux became weaker, as indicated by the slope of the curves, which become flatter with increased feed volume flow.

Figure 6 shows the influence of the cooling water flow on the transmembrane distillate flux.

With increasing volume flow of the cooling water, the distillate flux increased for almost all factor level combinations. It is noticeable that for many factor stage combinations, the increase in distillate flow became less evident with the increase of the cooling water volume flow.

#### 3.1.2. VMD

Figure 7, Figure 8, Figure 9 and Figure 10 show the results of the full factorial parameter variation.

The influence of the applied vacuum ∆p on the transmembrane distillate flux is shown in Figure 7.

Generally, the transmembrane distillate flux increased with increasing negative pressure. An exception was provided by the measuring series T_F1_, F_F1_, F_K1_, during which the distillate flux seemed to be unaffected by the applied pressure. The increase of the distillate flux was particularly pronounced across almost all factor levels at the lowest applied pressure of −0.15 bar.

Figure 8 depicts the distillate flux as a function of the feed temperature T_F_.

For all factor level combinations, the distillate flux increased linearly with increasing feed temperature, as the driving force—vapor pressure—increased.

Figure 9 depicts the transmembrane distillate flux as a function of the feed volume flow F_F_.

The increasing feed volume flow led to an increase in the transmembrane distillate flux, as the crossflow reduced the hydrodynamic boundary layer on the feed side of the membrane.

Figure 10 depicts the distillate flux as a function of the volume flow of the cooling water F_K_.

For all factor stage combinations, an increase in transmembrane distillate flux with increasing cooling water volume flow can be seen.

### 3.2. Long-Term Investigations

#### 3.2.1. AGMD

The temperatures of all solutions could be kept constant over time. During the investigation period, 63.1 kg of distillate were generated. This corresponds to a yield of ca. 49%. Based on the amount of distillate, an average distillate flux of 4.4 L/m^2^h was calculated. This is slightly lower than the value of 4.7 L/m^2^h achieved with the same parameters during the factor variation investigations.

Figure 11 depicts the resulting conductivity of feed/concentrate and distillate as a function of time.

The electrical conductivity of the feed increased linearly during the long-term investigation. Within the investigation period of 48 h, the conductivity almost doubled from 410.7 µS/cm^2^ to 806.3 µS/cm. The conductivity of the distillate increased over time and followed the conductivity of the feed. The initially low conductivity in the first hour did not reflect the performance of the membrane. Due to the low flow of both configurations, the conductivity sensor was mounted in a filling pipe. At the start of the long-term investigations, the pipe was filled with air. Contact with air at the beginning of the test runs influenced the sensor until it was fully covered with water. The rise and fall of electrical conductivity can be explained by the start-up performance of the MD plant and remaining solution in the membrane modules. The electrical conductivity of the distillate increased from, initially, 0.9 µS/cm to 2.3 µS/cm. Thus, during the entire test period, the distillate had an electrical conductivity, which corresponded to the quality of PW.

The calculated salt retention R during the AGMD long-term investigation is shown in Figure 12.

Basically, the salt retention R was permanently above 99.7%.

#### 3.2.2. VMD

The temperatures of all solutions could be kept constant over time. During the investigation period, 9.14 kg of distillate was generated. This corresponds to a yield of ca. 17.8%. Based on the amount of distillate, an average distillate flux of 3.05 L/m^2^h was calculated. This is slightly lower than the value of 3.16 L/m^2^h achieved with the same parameters during factor variation investigations.

Figure 13 depicts the resulting conductivity of feed/concentrate and distillate as a function of time.

The electrical conductivity of the feed increased linearly during the long-term investigation. Within the investigation period of 10 h, the conductivity increased from 435 µS/cm^2^ to 530 µS/cm. The conductivity of the distillate increased over time and followed the conductivity of the feed. The initially low conductivity can be explained by start-up procedures until the pipe that contained the sensor was filled with liquid. The electrical conductivity of the distillate increased shortly within the first hour, from, initially, 0.7 µS/cm to 1.6 µS/cm. Then, the conductivity decreased again, reached a value of 0.9 µS/cm and remained constant until the end of the investigation. Thus, during the entire test period, the distillate had an electrical conductivity, which corresponded to the quality of PW and WFI.

The salt retention R of the VMD as a function of the time is depicted in Figure 14.

The initially high salt rejection in the beginning of the long-term investigation is attributed to the insufficient construction details of the applied membrane plant, as described before. Salt retention was permanently above 99.6%. Decrease and increase in salt retention within the first five hours is justified by the course of the distillate conductivity, (Figure 12).

### 3.3. Microbiological Investigations

The results of the microbiological studies are presented in Table 7. In each case, two samples were taken from the feed water and from the distillate.

To determine the decontamination factor with regard to the total bacteria count, the respective results depicted in Table 5 were averaged. From these arithmetic average values, the decontamination factor for the two incubation temperatures was calculated. From this, the mean decontamination factor ∅DF was calculated. The decontamination factor for the total bacteria count is shown in Table 8.

About 4% of the total bacteria in the feed could be found in the distillate. Therefore, the MD provided a rejection of the microbiological impurities of approximately 96%. It is noticeable that the decontamination factor for the incubation at 22 °C is significantly lower than that for the incubation at 36 °C.

Table 9 depicts the total bacteria count estimated according to Ph. Eur.

The total bacteria count in the distillate met the requirements of PW but did not meet the requirements of WFI.

### 3.4. Material Stability

The drop test prior to and after the insertion of the membrane material showed that sterilization with an aqueous 2% H2O2 solution did not negatively affect the hydrophobicity of the membrane, as the measured contact angle was >90% in both conditions. The spacer and film material were also suitable for sterilization with the aforementioned solution.

## 4. Discussion

### 4.1. Parameter Variation

Table 10 depicts the effects of the feed temperature T_F_, feed volume flow F_F_ and cooling water flow F_K_ on the transmembrane distillate flux F_D_ during AGMD.

It is obvious that the feed temperature T_F_ is the decisive factor to increase the transmembrane distillate flow. The increase in the feed volume flow led to a reduction in the boundary layer resistance. However, this effect was limited. The effect of the cooling water flow F_K_ on the distillate flux was negligible.

Table 11 depicts the effects of the feed temperature T_F_, feed volume flow F_F_, cooling water flow F_K_ and applied negative pressure on the transmembrane distillate flux F_D_ during VMD.

Similar to what observed for AGMD, the feed temperature T_F_ had the greatest influence on the transmembrane distillate flux during VMD, followed by the feed volume flow F_F_ and the cooling water volume flow F_K_. The applied negative pressure ∆p had the smallest effect on the distillate volume flow. However, ∆p had to be kept above −0.2 bar, as the feed tank design did not allow lower values. Accordingly, the effect of negative pressure on the performance of VMD could not be predicted.

### 4.2. Long-Term Investigations

The long-term investigations showed that both AGMD and VMD were able to produce distillates with an electrical conductivity within the limiting values of PW.

Noticeable is the dependence of the electrical conductivity of the distillate on the conductivity of the feed during AGMD. It is assumed that an intrusion of the feed solution into the distillate cycle occurred. The intrusion of the feed was caused either by minimal defects of the sealings or by a too high pressure on the feed side of the membranes that caused mass transfer through the pores of the actually hydrophobic membranes (exceedance of the liquid entry pressure LEP). In addition, the investigations showed that a certain maximum concentration of the concentrate limited the process, as the quality of the distillate depended on it. The achievable yield of AGMD thus seems limited.

The long-term investigation of the VMD did not show a comparable behavior. The conductivity of the distillate did not increase when the concentration of the feed rose. However, VMD investigations were significantly shorter compared to AGMD ones, so the concentrate concentration did not reach a hypothetical critical value.

The achievable distillate fluxes were smaller compared to the respective fluxes measured during parameter variation. It is assumed that the reason is vapor pressure reduction due to the increasing salt concentration of the concentrate. The driving force of membrane distillation during the long-term investigations was reduced accordingly and led to the decrease of the distillate flux.

### 4.3. Microbiological Investigations

The total bacteria count in the distillate met the requirements of PW but not those of WFI. The measured bacteria counts were 30–40 times higher than the limit value of WFI and 250–330 times below the limit value of PW. However, neither the test plant nor the module fulfilled the demands of hygienic design, as the detachable connections in the product and distillate-contacting area were realized with screw connections.

## 5. Conclusions and Outlook

The investigations carried out showed that single-stage MD is suitable for the production of pharmaceutical-grade water. The generated distillate reached the claims of Ph. Eur. for PW regarding the electric conductivity and the total bacteria count.

The feed temperature was the decisive factor for the performance of both AGMD and VMD configurations. All other factors played a minor role. VMD showed no significant increase in transmembrane distillate flow with an applied vacuum. The main reason for this behavior is the limited negative pressure on the distillate side that could be achieved in the test facility. Future studies of VMD should therefore be carried out below −0.2 bar.

Future studies must also provide information on whether MD is comparable to distillation and membrane-assisted processes in terms of yield and specific energy demands. The decisive factor for this is the decoupling of the quality of the distillate from that of the feed.

In addition to the necessary technical investigations, the approval of the legislator is crucial for the establishment of MD as a possible alternative to distillation and cold production by RO in a strictly regulated market. Then, the decision of pharmaceutical manufacturers to use MD will depend on the strict adherence to operational safety. This also includes the approval of FDA or other institutions for any kind of material used in the plant.

Further applications of membrane distillation in the field of ultrapure water generation seem to be promising for the treatment of water for hydrogen production. The electrolysis used for water splitting requires a fully de-salted and softened feed water. Hygienic requirements similar to those of the pharmaceutical industry do not exist. Excess heat from the electrolysis plant can be used to heat the feed water.

## Figures and Tables

**Figure 1 ijerph-18-06058-f001:**
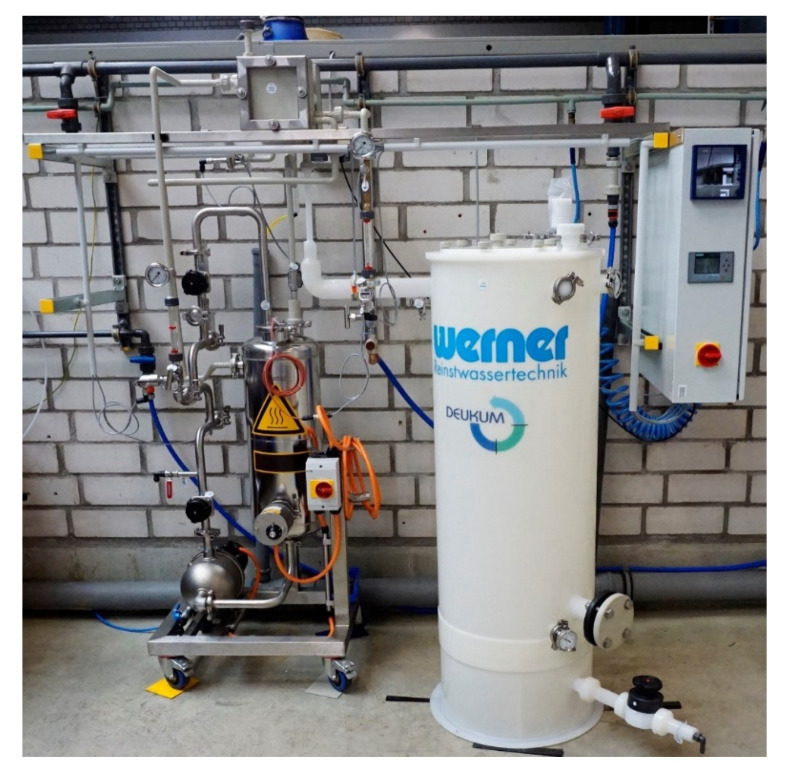
MD test plant.

**Figure 2 ijerph-18-06058-f002:**
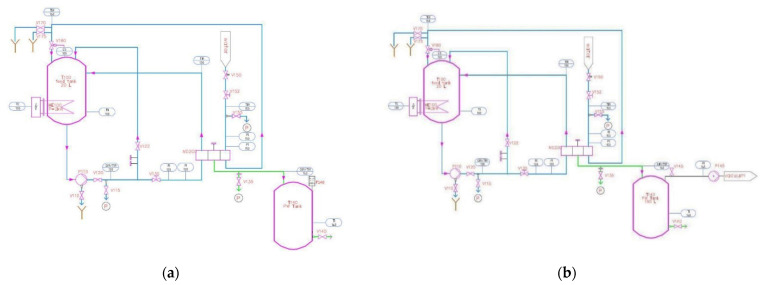
P + I schemes of the test plant. (**a**) AGMD configuration; (**b**) VMD configuration.

**Figure 3 ijerph-18-06058-f003:**
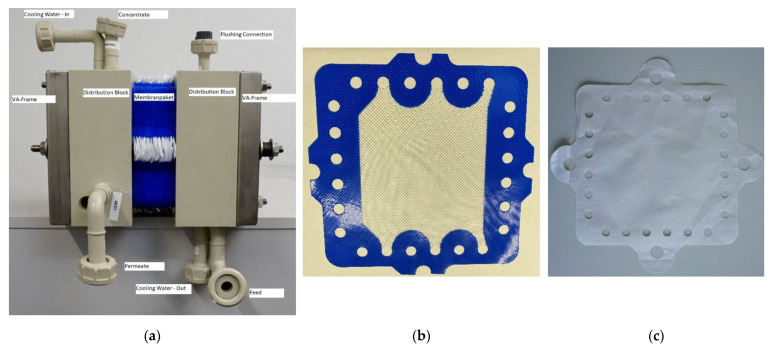
Membrane stack provided by Deukum. (**a**) General set-up; (**b**) single membrane spacer; (**c**) membrane.

**Figure 4 ijerph-18-06058-f004:**
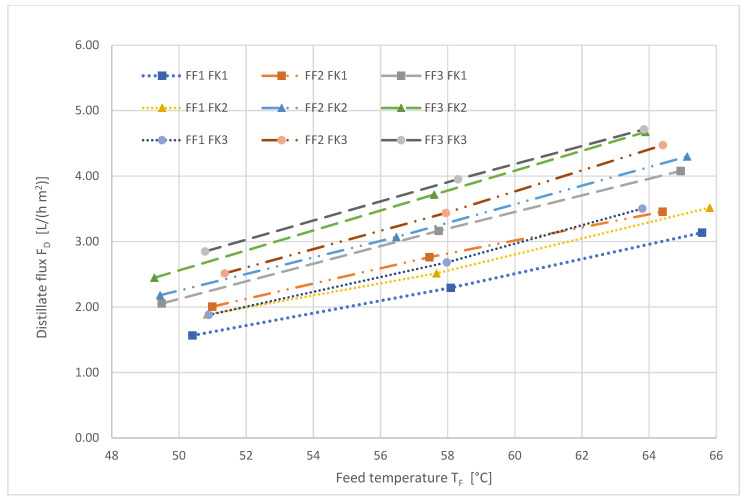
Distillate flux as a function of the feed temperature during AGMD at a constant cooling water temperature of 17.5 °C. The numbers refer to the different levels, low (1), medium (2) and high (3), described in Table 4.

**Figure 5 ijerph-18-06058-f005:**
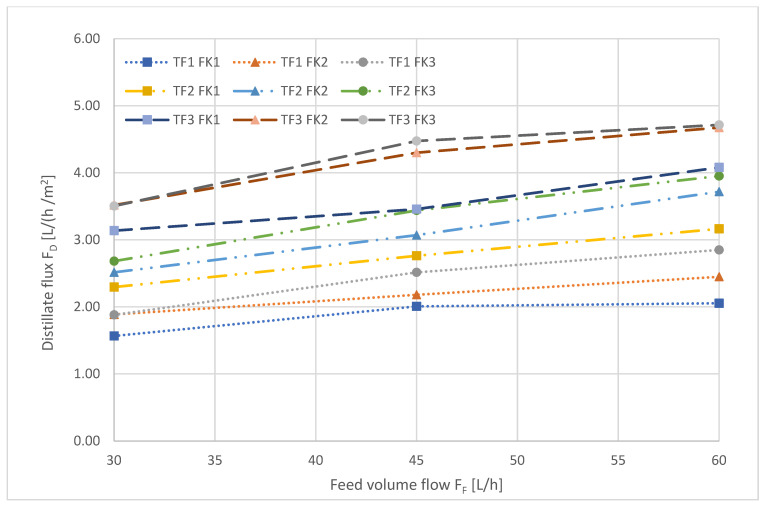
Distillate flux as a function of the feed volume flow (crossflow) during AGMD. The numbers refer to the different levels, low (1), medium (2), and high (3), described in Table 4.

**Figure 6 ijerph-18-06058-f006:**
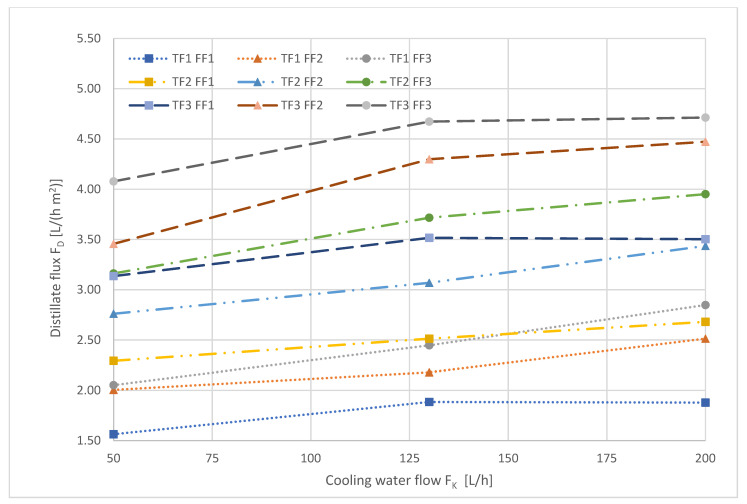
Distillate flux as a function of the cooling water flow. The numbers refer to the different levels, low (1), medium (2), and high (3), described in Table 4.

**Figure 7 ijerph-18-06058-f007:**
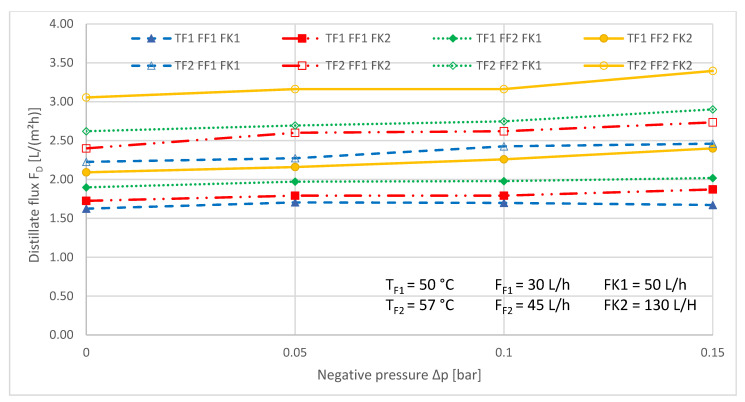
Distillate flux as a function of the applied negative pressure during VMD. T_F_ = const., F_F_ = const., F_K_ = const. during the respective measurement series. The numbers refer to the different levels, low (1) and medium (2), described in Table 4.

**Figure 8 ijerph-18-06058-f008:**
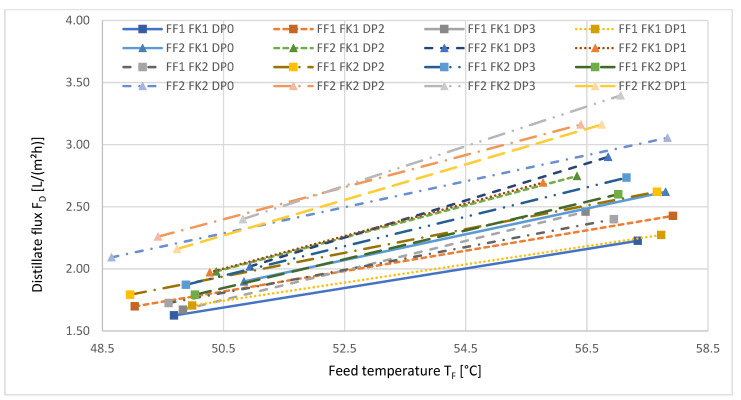
Distillate flux as a function of the feed temperature during VMD. Δp = const., F_F_ = const., F_K_ = const. during the respective measurement series. The numbers refer to the different reference levels, without pressure (0), low (1), medium (2), and high (3), described in Table 4.

**Figure 9 ijerph-18-06058-f009:**
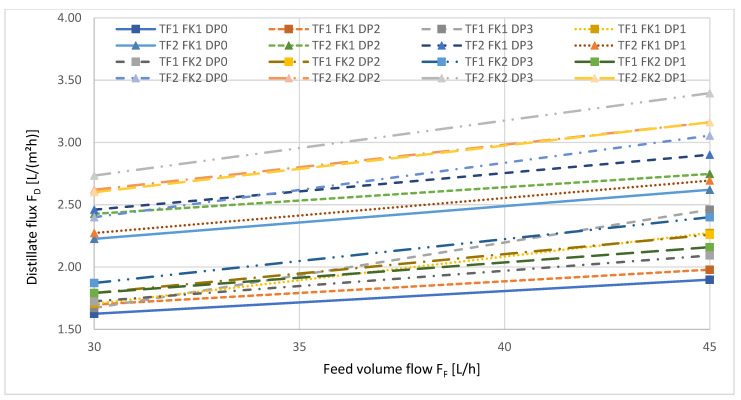
Distillate flux as a function of the feed volume flow (crossflow) during VMD. Δp = const., T_F_ = const., F_K_ = const. during the respective measurement series. The numbers refer to the different reference levels, without pressure (0), low (1), medium (2) and high (3), described in Table 4.

**Figure 10 ijerph-18-06058-f010:**
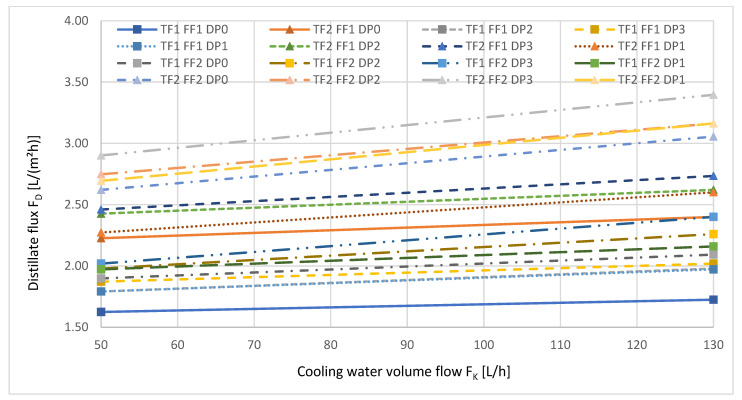
Distillate flux as a function of the volume flow of the cooling water F_K_ during VMD. Δp = const., F_F_ = const., T_K_ = const. during the respective measurement series. The numbers refer to the different reference levels, without pressure (0), low (1), medium (2) and high (3), reported in Table 4.

**Figure 11 ijerph-18-06058-f011:**
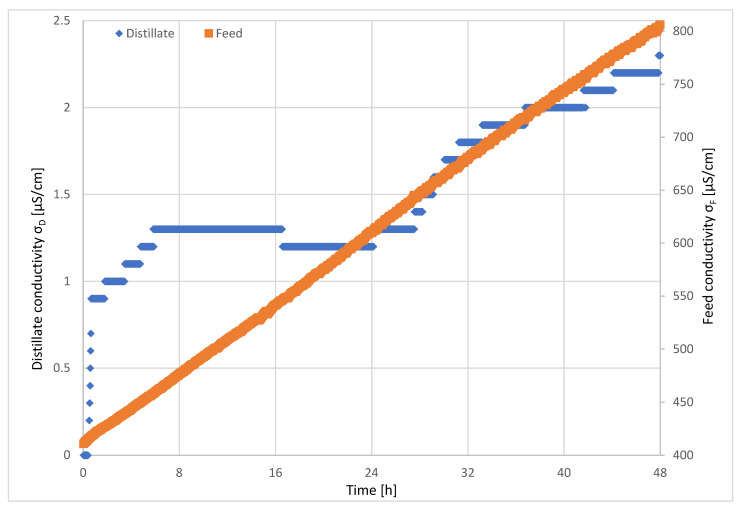
Feed and distillate conductivity during long-term investigations of AGMD. T_F_ = 64.6 °C, F_F_ = 60 L/h, F_K_ = 130 L/h, T_K_,_in_ = 16.9 °C.

**Figure 12 ijerph-18-06058-f012:**
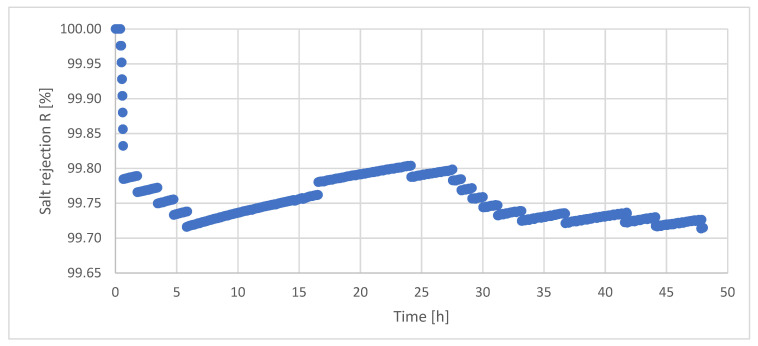
Salt rejection during long-term investigations of AGMD. T_F_ = 64.6 °C, F_F_ = 60 L/h, F_K_ = 130 L/h, T_K,in_ = 16.9 °C.

**Figure 13 ijerph-18-06058-f013:**
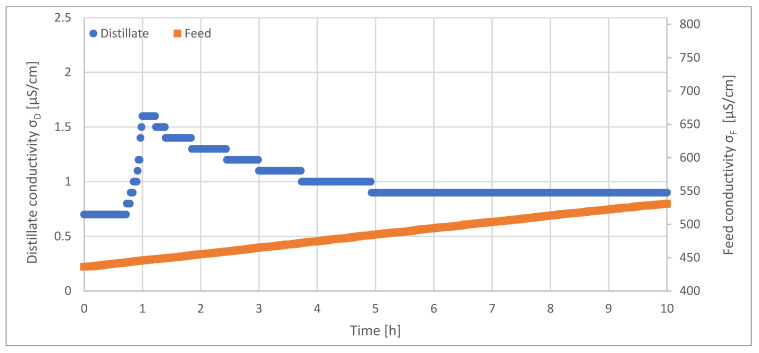
Feed and distillate conductivity during long-term investigations of VMD. T_F_ = 56.5 °C, F_F_ = 45 L/h, F_K_ = 130 L/h, T_K,in_ = 13.1 °C, Δp = −0.1 bar.

**Figure 14 ijerph-18-06058-f014:**
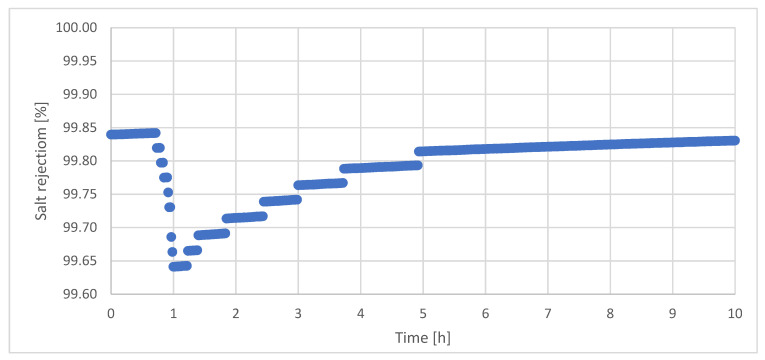
Salt rejection during long-term investigations of VMD. T_F_ = 56.5 °C, F_F_ = 45 L/h, F_K_ = 130 L/h, T_K,iin_ = 13.1 °C, Δp = −0.1 bar.

**Table 1 ijerph-18-06058-t001:** Limiting values for PW and WFI [2,3].

Parameter	Limiting Values PW	Limiting Values WFI
Conductivity @ 20 °C	4.3 µS/cm	1.1 µS/cm
Nitrate	≤0.2 ppm	≤0.2 ppm
Heavy metals	≤0.1 ppm	
Total organic carbon TOC	≤0.5 mg/L	≤500 ppb
Bacteria endotoxins		≤0.25 IU/mL ^2^
Aerobic germs and bacteria	≤10,000 CFU/100 mL ^1^	≤10 CFU/100 mL

^1^ CFU colony-forming unit. ^2^ IU international unit.

**Table 2 ijerph-18-06058-t002:** Characteristics of common MD module configurations [14,15].

Configuration	Characteristics	Advantages	Disadvantages
DCMD	Both sides of the membrane are in contact with a liquid. The extraction of the condensation enthalpy of the vapor takes place on the permeate side in cooling water	Simple operation Simple plant setup Simple module setup High flux	Compared to other configurations: Highest heat losses caused by conduction through the membranes Not suitable for the removal of non-volatile organics
AGMD	The feed side of the membrane is in contact with a liquid, the permeate side is in contact with a stagnant gas layer The condensation of the distillate takes place on a separate dense surface (condensation film) adjacent to the air gap, which is cooled by cooling water	Low conductive heat losses Low fouling tendency No wetting on the permeate side of the membrane	Additional resistance to mass transfer Complicated module design Low flux
SGMD	The feed side of the membrane is in contact with a liquid, the permeate side is in contact with a sweep gas stream (e.g., air) Water vapor condensation takes place outside the module	No wetting on the permeate side of the membrane Low thermal polarisation	Large condenser required (small amount of permeate in a large volume of sweep gas) Low flux
VMD	The feed side of the membrane is in contact with a liquid, the permeate side is under negative pressure Water vapor condensation takes place outside the module	High flux Little conductive heat losses	Higher risk of pore wetting Vacuum pump and external condenser required

**Table 3 ijerph-18-06058-t003:** Membrane characteristics. Product Datasheet Lydall Performance Materials, “14P02F: Microporous Polyethylene Film”.

Parameter	Property
Material	Polyethylene
Nominal cut-off	0.2 µm
Porosity	85%
Membrane thickness	110 µm

**Table 4 ijerph-18-06058-t004:** Conditions of factor variation investigations.

Configuration	Level	T_F_ [°C]	F_F_ [L/h]	F_K_ [L/h]	∆p [bar]
AGMD	Low (1)	50 ± 2	30	50	0
	Medium (2)	57 ± 2	45	130	0
	High (3)	65 ± 2	60	200	0
VMD	Reference (0)				0
	Low (1)	50 ± 2	30	50	−0.05
	Medium (2)	57 ± 2	45	130	−0.1
	High (3)				−0.15

**Table 5 ijerph-18-06058-t005:** Operating conditions of long-term investigations.

Parameter	AGMD	VMD
Feed temperature T_F_	65 ± 2 °C	57.5 ± 2 °C
Feed volume flow F_F_	60 L/h	45 L/h
Cooling water volume flow F_K_	130 L/h	130 L/h
Negative pressure ∆p	-	−0.1 bar
Duration	48 h	10 h

**Table 6 ijerph-18-06058-t006:** Operating conditions prior to microbiological sampling.

Parameter	Property
Feed temperature T_F_	65 ± 2 °C
Feed volume flow F_F_	45 L/h
Cooling water volume flow F_K_	130 L/h

**Table 7 ijerph-18-06058-t007:** Total bacteria count according to TVO, (CFU/mL).

Temperature	Feed, Sample 1	Feed, Sample 2	Distillate, Sample 1	Distillate, Sample 2
22 °C	>300	>300	7	3
36 °C	120	130	9	7

**Table 8 ijerph-18-06058-t008:** Decontamination factor.

Incubation Temperature	DF
22 °C	1.67%
36 °C	6.40%
∅DF	4.04%

**Table 9 ijerph-18-06058-t009:** Total bacteria count according to Ph. Eur.

Sample	Total Bacteria Count [CFU/mL]
Distillate, sample 1	4
Distillate, sample 2	3

**Table 10 ijerph-18-06058-t010:** Effects of the parameters on the transmembrane distillate flux F_D_ during AGMD.

Parameter	$\frac{F_{D}}{A_{M}}\left[\frac{L}{m^{2}\cdot h}\right]$	$\frac{∆F_{D}/A_{M}}{F_{D}/A_{M}}\left[\%\right]$	$\frac{∆factor}{factor}\left[\%\right]$	$\frac{∆F_{D}/A_{M}}{F_{D}/A_{M}}/\frac{∆factor}{factor}$
T_F_	1.83	85	30	2.83
F_F_	0.96	38	100	0.38
F_K_	0.61	22	300	0.07

**Table 11 ijerph-18-06058-t011:** Effects of the parameters on the transmembrane distillate flux F_D_ during VMD.

Parameter	$\frac{F_{D}}{A_{M}}\left[\frac{L}{m^{2}\cdot h}\right]$	$\frac{∆F_{D}/A_{M}}{F_{D}/A_{M}}\left[\%\right]$	$\frac{∆factor}{factor}\left[\%\right]$	$\frac{∆F_{D}/A_{M}}{F_{D}/A_{M}}/\frac{∆factor}{factor}$
∆p	0.23	10	200	0.05
T_F_	0.85	42	15	2.8
F_F_	0.48	23	50	0.46
F_K_	0.28	13	160	0.08

## Data Availability

Data are contained within the article.

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
