# Peer review of "Membrane Distillation for the Production of Pharmaceutical-Grade Water—Investigation into the Application of AGMD and VMD"

_ijerph, 2021, doi:10.3390/ijerph18116058_

Round 1
Reviewer 1 Report
The production of pharmaceutical-grade water is an important and significant task, which was solved by the Authors excellently, however, there are some problems in the manuscript that should be corrected or answered.
- The type of the legend of the Tables is not the same: in the case of Tables 3., 5., 6 are italics, in the others not
- Line 105 and line 106: tab(!) water; it should be: tap water
- Various explanations of symbols are missing in the figures: Figures 5-11, FF1FK1, TF1FK1 etc. Please make up for the legend
- 9-11: the lines are defined by only two points. Why?
- Line 147 and line 162: 2 wt.-%, „.-„ is not necessary
- Line 54: und: it should be: and
- Line 67: after is not necessary at all
- References are confused, bold and italics…, please correct the parameters of references.
Author Response
Dear reviewer,
please see the attachment.
Best regards,
Frank Rögener

Reviewer 2 Report
Water purification using two different membrane-based models has been investigated. The subject is interesting and valuable. However, there are some points which should be considered by the authors before final publication. I suggest revision of the manuscript based on the following points:
- The abbreviation of TOC in Table 1 should be defined.
- Table 1 can be also further completed by ORP values, which is an important parameter for purified water.
- Concerning the MD, it has been stated that “When drinking water is used as a feed solution, these membranes allow the passage of water vapor only and retain suspended and dissolved matter on the retentate side; thus the condensed product obtained is theoretically 100% pure water.”. However, the recent theoretical-based reports which try to consider the required conditions of realization, predict the purities lower than 100% (see, for example, [The Journal of Physical Chemistry B 123 (2019) 8740-8752] & [Nano Lett.2012, 12, 7, 3602–3608] for graphene-based membranes, as new ones). This statement (100% efficiency in the theoretical studies) should be improved in the introduction section by mentioning some recent studies.
- The material of the membrane is polyethylene. Hence, could the authors comment about the polyethylene molecule release into the purified water?
- All the data points given in the figure need error bar. The origin of the dispersion (statistical and/or technical dispersions) should be mentioned.
- Figure 12 shows that the obtained distilled water is not suitable for the WFI only after the short time of ~1h. Is that an expectable result? Please compare it with the results given in the literature. Also, discuss about this fast change in characteristic of the distilled water in the AGMD and a reverse behavior in the VMD (Figure 14).
- In Figure 13, the initial salt rejection has been presented as well as 100%. But, as mentioned in the previous comments, it cannot be right even in theoretical models trying to approach into the reality. This should be clarified by the authors.
Author Response

(The authors gave the same response as above.)

Round 2
Reviewer 1 Report
-
Reviewer 2 Report
The manuscript has been revised based on the comments and now it is publishable.